# Chronic Neuronal Hyperexcitation Exacerbates Tau Propagation in a Mouse Model of Tauopathy

**DOI:** 10.3390/ijms25169004

**Published:** 2024-08-19

**Authors:** Itaru Nishida, Kaoru Yamada, Asami Sakamoto, Tomoko Wakabayashi, Takeshi Iwatsubo

**Affiliations:** 1Department of Neuropathology, Graduate School of Medicine, The University of Tokyo, Tokyo 1130033, Japan; itaru.nishida826@gmail.com (I.N.); sakamoto@m.u-tokyo.ac.jp (A.S.); t-wakabayashi@my-pharm.ac.jp (T.W.); 2Department of Pathophysiology, Meiji Pharmaceutical University, Tokyo 2040004, Japan; 3National Institute of Neuroscience, National Center of Neurology and Psychiatry, Tokyo 1878551, Japan; iwatsubo@m.u-tokyo.ac.jp

**Keywords:** Alzheimer’s disease, tau, propagation, neuronal activity, entorhinal cortex, hippocampus

## Abstract

The intracerebral spread of tau is a critical mechanism associated with functional decline in Alzheimer’s disease (AD) and other tauopathies. Recently, a hypothesis has emerged suggesting that tau propagation is linked to functional neuronal connections, specifically driven by neuronal hyperactivity. However, experimental validation of this hypothesis remains limited. In this study, we investigated how tau propagation from the entorhinal cortex to the hippocampus, the neuronal circuit most susceptible to tau pathology in AD, is affected by the selective stimulation of neuronal activity along this circuit. Using a mouse model of seed-induced propagation combined with optogenetics, we found that the chronic stimulation of this neuronal connection over a 4-week period resulted in a significant increase in insoluble tau accumulation in both the entorhinal cortex and hippocampus. Importantly, the ratio of tau accumulation in the hippocampus relative to that in the entorhinal cortex, serving as an indicator of transcellular spreading, was significantly higher in mice subjected to chronic stimulation. These results support the notion that abnormal neuronal activity promotes tau propagation, thereby implicating it in the progression of tauopathy.

## 1. Introduction

Tau is a key protein involved in neuronal death in a range of neurodegenerative diseases, most notably Alzheimer’s disease (AD). Extensive histopathologic observations by Braak et al. have long established a distinct spatiotemporal pattern in the progression of tau pathology in AD brains, originating in the entorhinal cortex and extending to the hippocampus, then to the adjacent temporal and limbic cortices [1]. The stereotypical progression of tau pathology, known as the Braak stage, of postmortem brains has also been confirmed via tau Positron Emission Tomography (PET) imaging in living individuals [2], and it strongly predicts the rate of future atrophy [3,4].

The molecular basis of the spread of tau to different brain regions is thought to involve the cell-to-cell propagation of tau aggregates. Evidence produced over the past decade has suggested that tau not only aggregates cell-autonomously within neurons but also spreads between neurons in a manner similar to prion-like mechanisms [5,6]. Once released from neurons, tau aggregates are taken up by other neurons and serve as templates for recruiting native tau and facilitating further aggregation.

This propagation of tau aggregates occurs along neuronal circuits in both mouse models and human brains [7,8]. In addition to structural connections, the importance of functional connections in tau propagation was recently highlighted by a study using magnetoencephalography combined with 18F-flortaucipir PET [9]. Another study identified genes whose expression correlates with the spatial spread of tau pathology and demonstrated that glutamatergic synaptic genes influence tau propagation [10]. The discovery of tau with a higher propensity to aggregate, referred to as “seeding activity” or oligomeric tau in synapses of AD brains, also suggests that tau spreads along synaptically connected brain regions [11,12].

Given that cortical and hippocampal hyperactivity is a hallmark of early-stage AD [13,14], these studies suggest that neuronal activity, particularly in highly functionally connected regions, is a robust predictor of the propagation of tau pathology in the human brain.

Studies using cellular and animal models to explore the interplay between neuronal activity and tau pathophysiology have yielded several intriguing findings. For example, soluble tau is released from cells, and this process is facilitated by increased activity [15,16]. Tau translocation in hippocampal neurons cultured in microfluidic devices was shown to be promoted by synaptic connections [17]. In addition, Wu et al. showed that increasing neuronal activity in tau transgenic mice rTg4510 leads to increased local tau pathology [18]. 

These findings suggest a profound link between tau propagation and neuronal activity. Despite these advances, many previous studies have not clearly distinguished and examined the effect of neuronal hyperactivity on cell-autonomous aggregation versus tau propagation, especially in vivo or inert/soluble tau versus aggregated/insoluble tau, leaving several critical questions unanswered. Does neuronal hyperactivity promote local tau aggregation or the spread of tau to different brain regions? Does neuronal hyperactivity promote the propagation of inert/soluble tau or aggregated tau? To address these knowledge gaps, we sought to determine whether neuronal hyperexcitability enhances the spread of aggregated tau pathology in a mouse model of seed-induced propagation combined with optogenetics.

## 2. Results

### 2.1. Tau Pathology Propagation from EC to Hippocampus Following Tau Fibril Microinjection

The entorhinal cortex (EC) serves as one of the initial sites where tau pathology accumulates in AD brains and provides the major input to the hippocampus. To mimic the progression of tau pathology in AD, where tau pathology spreads from the EC to the hippocampus, we performed unilateral microinjections of recombinant tau fibrils into the left medial entorhinal cortex of 2.5-month-old P301S human tau transgenic mice (PS19) (Figure 1A). PS19 mice are known to exhibit more robust tau propagation across different brain regions compared to wildtype mice upon fibril injection; therefore, we decided to utilize PS19 mice for this experiment. At 3.5 months (14 weeks) post-injection, when the mice were 6 months old, we performed immunohistochemistry to determine whether tau pathology was spreading in the hippocampus in the coronal brain sections (Figure 1A). To control for any effects of the injection procedure, we injected phosphate-buffered saline (PBS) as a control. Tau pathology in PS19 mice does not occur until after 6.5 months of age without fibril injections. Consistent with this, immunostaining of brain sections with the anti-phosphorylated tau antibody PHF1 revealed no tau pathology in either the EC injection site or the hippocampal regions in the PBS-injected control PS19 mice (Figure 1B). In contrast, in mice injected with tau fibrils, tangle-like structures in the EC and the CA1 region hippocampus were stained with PHF-1. The injected tau fibrils consist solely of the tau repeat domain and lack the epitope of PHF-1; therefore, this staining indicates newly aggregated tau induced by the injected fibrils. In contrast to CA1, other areas of the hippocampus, such as CA3 and the dentate gyrus (DG), showed minimal tau accumulation. No obvious behavior alterations associated with seizure were observed in the tau injection group compared to the control. 

### 2.2. Immunohistochemical Analysis of the Effects of Chronic Neuronal Stimulation on Tau Propagation

Next, we applied optogenetics to the seed-induced propagation model to study the effects of chronic neuronal stimulation on tau pathology in the EC and hippocampus. For chronic stimulation of neuronal activity, we used a mutant form of Channlerhodopsin2 (ChR2) containing two point mutations (C128S and D156A), taking advantage of its unusual ability to increase the probability of spiking to endogenous synaptic inputs for up to 30 min via a single instance of light stimulation [19,20]. We used adeno-associated virus vectors (AAV1) expressing ChR2^C128S/D156A^ fused to eYFP at the C terminus under a CaM kinase II promoter or AAV1-expressing eYFP as a control (Figure 2A). We unilaterally injected the above AAVs into the EC of 2.5-month-old PS19 mice simultaneously with recombinant tau fibrils to induce the propagation of tau to the hippocampus (Figure 2A). Starting 4 weeks later, we delivered daily 2 s pulses of blue light through optical cannulas implanted in the EC once a day for 4 weeks. At the end of the 4-week stimulation, when the mice reached 4.5 months of age, we first verified whether chronic light stimulation effectively activated neural activity in the EC and hippocampus by immunostaining for c-fos, one of the immediate early genes and frequently used as an indirect molecular marker of neuronal activity (Figure 2B). In the eYFP group, c-fos-positive neurons were sparse, indicating that they reflected intrinsic neuronal activity. In the ChR2^C128S/D156A^ group, however, significant upregulation of c-fos was observed not only in the EC but also in the granule cell layer of DG, the pyramidal cell layer of CA3, and the CA1 regions of the hippocampus, suggesting successful enhancement of neural activity in the chronic stimulation paradigm (Figure 2B). In addition, although the staining intensity was lower, tau accumulation was observed in corresponding regions of the contralateral hemisphere to those injected, showing a similar pattern (Appendix A). Increases in c-fos positive cells were also observed in contralateral EC and the hippocampus (Appendix A).

We then performed immunostaining for phosphorylated tau (p-tau) using the PHF-1 antibody to investigate how tau propagation from the EC to the hippocampus is altered by chronic hyperactivity. In the eYFP group, consistent with the data described in Figure 1B, a small number of tangle-like accumulations were observed predominantly in the EC and hippocampal CA1 region (Figure 2C). In the ChR2^C128S/D156A^ group, there was an increase in tangle-like tau accumulation in the cell bodies in the EC and the hippocampal CA1 compared to the eYFP group (Figure 2C,D). The increase in tau deposition was also observed in the rostral hippocampus, ~1 mm away from the EC in the anterior–posterior direction, indicating that tau accumulation spreads to distant hippocampal regions. No p-tau staining was observed in PS19 mice subjected to chronic stimulation without fibril injection, suggesting that chronic stimulation promotes seed-dependent tau propagation (Appendix A). 

We reasoned that if chronic neuronal hyperactivation facilitates the spread of tau pathology from the EC to the hippocampus, then there should be an increase in tau accumulation in the hippocampus beyond the effect of increased tau accumulation observed in EC [21]. Consistent with this idea, the ratio of tau accumulation in the hippocampus to tau accumulation in the EC for each mouse (propagation index) was significantly increased in the ChR2^C128S/D156A^ group (Figure 2E). Severe atrophy accompanied by brain volume loss was not observed in the eYFP group and the ChR2^C128S/D156A^ group.

### 2.3. Biochemical Analysis of the Effects of Chronic Neuronal Stimulation on Tau Propagation

Next, to confirm whether the observed increase in tau pathology was accompanied by an increase in aggregated tau, we repeated an experimental scheme similar to the immunohistochemical analysis but extended the duration of chronic stimulation from 4 weeks to 6 weeks using a different cohort of mice and performed biochemical experiments. Brains were subjected to serial extractions and tau levels were quantified in RIPA-soluble fractions (hereafter referred to as soluble fractions) and RIPA-insoluble and SDS soluble fractions (hereafter referred to as insoluble fractions) were quantified. Although tau was not detected at all in the insoluble fractions from young PS19 mice that did not undergo fibril injection (Appendix A), tau bands positive for both human tau and phosphorylated tau antibodies were evident in the insoluble fractions from both the eYFP and ChR2^C128S/D156A^ groups. In both immunoblot and ELISA, insoluble tau showed an increase in the ChR2^C128S/D156A^ group compared to the eYFP group, supporting the notion that chronic neuronal hyperactivity increases insoluble tau accumulation (Figure 3A–C). Interestingly, when normalizing p-tau with total tau (p-tau ratio), the value in the hippocampus of the ChR2 ^C128S/D156A^ group was found to be lower than that in the eYFP group, although there were no changes in the p-tau ratio in the EC of either groups. This suggests that the insoluble tau propagated to the hippocampus due to hyperexcitability has a decreased level of phosphorylation. In contrast to insoluble fractions, the soluble fraction of tau showed an increase in immunoblot analysis but a decrease in ELISA (Figure 4A–C). In the ELISA assay without SDS dissociation, it is possible that molecules such as oligomers are not detected due to epitope masking. Therefore, this result may reflect a potential decrease in monomeric tau and a concomitant increase in SDS-dissociable oligomeric tau. The p-tau ratio in the soluble fraction remained unchanged between eYFP and ChR2^C128S/D156A^ groups, suggesting that chronic neuronal stimulation did not solely facilitate the phosphorylation of tau.

## 3. Discussion

In this study, we investigated the effects of chronic neuronal excitation on tau propagation using a combination of optogenetics and a seed-induced model of tau propagation. Our results show that stimulation of the neural circuit from the EC to the hippocampus exacerbates tau pathology in both brain regions. Notably, the mice subjected to chronic stimulation exhibited even greater tau accumulation in the hippocampus compared to the increase observed in the EC. These results suggest that chronic stimulation not only exacerbates local tau deposition but also promotes the trans-regional spread of tau pathology. These findings strongly support the idea that neuronal hyperactivity promotes the spread of tau aggregates, thereby implicating it in the progression of tauopathy.

Recent studies have shown that not only insoluble/aggregated tau but also inert soluble tau are released extracellularly, undergo cellular uptake, and shuttle between cells [22,23]. Interestingly, it has been suggested that the intercellular transfer of AAV-expressed soluble tau also increases with neuronal activity [24]. Our study, among others, underscores the importance of elucidating whether the propagation of soluble and aggregated tau, both activated by neuronal activity, occurs via common molecular mechanisms or divergent downstream pathways to halt aggregated tau propagation without compromising its normal function.

From a therapeutic perspective, it is critical to understand the molecular mechanisms behind the enhanced impact of chronic neuronal activity on tau propagation. Mechanisms related to tau propagation, such as the release of seed-competent tau, may be enhanced in response to increased neuronal activity. Indeed, secretion via lysosomal exocytosis or exosomes, both of which have been implicated in the release of seed-competent tau [25,26], has been shown to occur in response to neuronal activity [27,28]. In addition, increased neuronal activity may enhance tau propagation by affecting the intracellular uptake or aggregation process of tau. Some endocytic processes known to be upregulated in response to activity [29] may also play a role in the uptake of seed-competent tau. The precise influence of neuronal activity on tau aggregation remains uncertain, but it is plausible that mechanisms such as increased local translation induced by neuronal activity could influence the aggregation process of tau. This possibility is consistent with the increase in tau observed in the RIPA fraction of the ChR2^C128S/D156A^ group. 

In our tau fibril injection experiment, in the EC, tau pathology was predominantly observed in CA1, while tau accumulation in regions such as CA3 and DG was sparse. In AD brains, tau pathology spreads to CA1 after its initiation in EC and does not appear in DG until the advanced stages of AD [30]. However, previous studies have typically used a mouse model in which tau spreads from the EC to the DG [31]. The robust accumulation of tau in CA1 in this study suggests the possibility that this hippocampal subregion may be more susceptible to tau propagation or aggregation. This finding is consistent with the results of tau propagation from Wolframin-1-positive EC neurons to CA1, as recently reported [32]. We demonstrated that hyperexcitability exacerbates tau propagation, but whether this leads to functional impairment or cell death remains unclear. A behavioral assay, particularly one assessing memory function, along with molecular assays to evaluate cell or synaptic loss, will be important in future studies to further investigate the role of activity-dependent tau propagation.

While the PS19 model is widely used to study tau pathology, it may not fully capture all aspects of AD pathology. Thus, using this model with P301S tau mutation and lacking amyloid pathology may limit the direct translatability of our findings to AD. Another limitation of this study is the duration of activity stimulation. In mouse models where tau propagation occurs over weeks to several months, it is imperative to perform chronic activity stimulation experiments over the same time period to accurately assess the effects of neural hyperactivity. Although the magnitude of neuronal activity elicited in our chronic stimulation paradigm should be ideally determined by electrical recording, this prolonged stimulation resulted in intense neuronal excitation, a phenomenon not typically observed in acute activity stimulation paradigms. Indeed, during our chronic stimulation, various behavioral changes were observed, ranging from severe seizures to milder phenotypes such as immobility, as we have previously described [20]. It is worth noting the frequent co-occurrence of epilepsy with AD, where the risk of seizures escalates with disease duration [33]. Furthermore, increased phosphorylated tau pathology is observed across many seizure disorders, indicating the potential causal link between seizure and tau pathology [34,35]. Nonetheless, future studies using more moderate stimulation paradigms are essential to meticulously elucidate how different levels of neuronal hyperactivity influence tau propagation.

In conclusion, this study provides evidence that neuronal hyperactivation exacerbates trans-regional tau propagation. A close relationship between tau and hyperexcitability has been suggested [36,37]. Our findings suggest the existence of a positive feedback loop where tau-induced hyperexcitability could further accelerate tau propagation. Interestingly, using the same optogenetic experiments with ChR2^C128S/D156A^, we have previously shown that neuronal hyperexcitability also increases the accumulation of Aβ [20]. This suggests that molecular events leading to hyperexcitability may be potential targets for therapeutic intervention in AD. Further investigation is warranted to elucidate how neuronal hyperexcitability exacerbates tau aggregate propagation and how it differs from the mechanisms of normal tau transfer.

## 4. Materials and Methods

### 4.1. Mice

Both male and female mice of PS19 mice over-expressing P301S mutant tau (stock number: 008169 Jackson laboratory) [38] were used for the experiments. Tau knock-out mice were obtained from The Jackson Laboratory (Bar Harbor, ME, USA). Wildtype mice on C57B6/J background were purchased from Japan SLC (Shizuoka, Japan).

### 4.2. Recombinant Tau Purification and Fibrilization

Recombinant tau repeat domain (RD) with P301S mutation was purified as previously described [26]. Purified recombinant RD/P301S tau was diluted in 30 mM Tris-HCl (pH 7.5), followed by the addition of Heparin (Thermo Fisher Scientific Inc., Waltham, MA, USA) to a final concentration of 0.1 mg/mL, 10 mM DTT, 0.1% Sodium azide, and cOmplete (Roche, Mannheim, Germany). The tau fibrillization reaction was conducted at 37 °C for 96 h. After the fibrillization reaction, centrifugation was carried out at 113,000× *g*, 4 °C, for 20 min, and the tau fibrils were sedimented. After sedimentation, PBS was added, and centrifugation was repeated at 113,000× *g*, 4 °C, for 20 min to wash out the remaining monomer. Subsequently, the fibrils were suspended in PBS and subjected to sonication on ice until the solution became clear, approximately 30 s, to break the fibrils.

### 4.3. Stereotaxic Injection of AAV and Tau Fibrils

AAV1 vectors expressing ChR2^C128S/D156A^-eYFP (3.5 × 10^13^ genome copy/mL) [19,20] or eYFP (3.4 × 10^13^ genome copy/mL) were obtained from Penn Vector core (Philadelphia, PA, USA). Mice were anesthetized and secured onto a brain stereotaxic frame (Kopf instruments, Tujunga, CA, USA). After shaving the head and making a midline incision to expose the skull, the coordinates for the medial–lateral (M/L) and dorsal–ventral (D/V) directions relative to bregma and lambda were adjusted to within a ±0.01 mm difference. Subsequently, the antero–posterior (A/P) direction was adjusted to position the tip of the dental drill at coordinates 4.75 mm anterior and 3.00 mm left lateral from bregma to target medial EC, while being careful not to damage the brain surface when using a dental drill aided by a stereomicroscope (Evident, Tokyo, Japan). A glass capillary with an outer diameter of approximately 100–120 µm was connected to a 10 µL Hamilton syringe (Hamilton company, Reno, NV, USA) to aspirate the injection material into the capillary. After setting the coordinates again to 4.75 mm anterior and 3.00 mm left lateral from bregma, the capillary was slowly inserted into the brain to a depth of 3.70 mm ventrally, and the material was injected into the brain at a rate of 0.1 µL/min using a syringe pump (Kd Scientific, Holliston, MA, USA). For the injection of tau fibrils only, 5.0 µg/2 µL of the tau fibril solution prepared was injected per mouse. For the injection of AAV1 virus only, 0.5 µL was injected per mouse. For chronic stimulation experiments requiring the simultaneous microinjection of virus and tau fibrils into the brain, the tau fibril solution and virus solution were mixed at a ratio of 4:1 before filling the glass capillary and injecting 2.5 µL per mouse (equivalent to 5.0 µg of tau fibrils and 0.5 µL of AAV1 virus solution). After the completion of injection, the capillary was slowly withdrawn after a 10 min pause to prevent the reflux of the injected material. The mouse scalp was sutured with nylon thread.

### 4.4. Optogenetics

Three weeks after the virus injection, the mice were anesthetized again, and surgery was performed to insert an optic cannula. A hole was made with a dental drill at the target coordinates (A/P: −4.75 mm, M/L: 3.00 mm), and an optic cannula (Doric Lenses, Quebec City, QC, Canada) was then mounted and slowly inserted to a point 3.45 mm ventrally from the brain surface. To prevent the optic cannula from detaching, a bone screw (BASi, West Lafayette, IN, USA) was inserted into the frontal part of the right hemisphere as an anchor. The optic cannula and bone screw were secured with dental cement (SUN MEDICAL, Shiga, Japan). 

To conduct optical stimulation, a 473 nm blue laser (Shanghai Dream Lasers Technology Co., Ltd., Shanghai, China) was connected to the optic cannula implanted in the mouse brain via an optical fiber patch cord (Doric Lenses, Quebec City, QC, Canada). Four weeks after the virus injection, blue light stimulation at 473 nm and 4 mW intensity was applied for 2 s/day for 28 days through optic cannulas implanted in the medial EC once a day for 4 weeks for immunohistological analysis and for 6 weeks for biochemical analysis. On the final day, the mice were anesthetized, and brains were removed 90 min later from blue light stimulation.

### 4.5. Immunohistochemistry

Brains were fixed in 4% PFA/PBS for 24 h, embedded in paraffin blocks, and sectioned coronally at 4 μm thickness. For antigen retrieval, de-paraffinized sections were treated with microwave radiation (550 W, 10 min) in citrate buffer (pH 6.0) prior to immunostaining. For staining with PHF-1, sections were further treated with 100 µg/mL proteinase K for 6 min at 37 °C prior to PHF-1 staining (a kind gift from Dr. Peter Davies) [39]. For eYFP staining, de-paraffinized sections were treated with microwave radiation (550 W, 10 min) in 100 mM Tris-HCl buffer (pH 9.2) prior to immunostaining. Some brain sections underwent counterstaining with hematoxylin and eosin in addition to immunostaining to visualize the CNS architecture. Mice with significantly low YFP staining showed extremely low PHF1 staining as well, suggesting that not only the AAV1 virus but also the simultaneously administered tau fibers were not sufficiently injected into these individuals. For this reason, such individuals were excluded from further analysis.

### 4.6. Biochemistry

Serial brain extraction was performed with a modified protocol from a previous study [7]. Dissected cortices and hippocampus were homogenized with RIPA buffer (50 mM Tris, 150 mM NaCl, 0.1% SDS, 0.5% sodium deoxycholate, 1% NP-40, 5 mM EDTA, 1mM PMSF, pH = 8.0) containing cOmplete^TM^ protease inhibitor cocktails (Roche) and PhosSTOP^TM^ (Roche) and ultracentrifuged at 100,000× *g* for 30 min at 4 °C. Supernatants were saved as 1st RIPA soluble fractions (RIPA1) and stored at −80 °C. The resulting pellets were homogenized again with RIPA buffer containing cOmplete^TM^ protease inhibitor cocktails (Roche) and PhosSTOP^TM^ (Roche) and ultracentrifuged at 100,000× *g* for 30 min at 4 °C. Supernatants were saved as 2nd RIPA soluble fractions (RIPA2) and stored at −80 °C. RIPA insoluble pellets were homogenized with 1M sucrose in RIPA buffer and ultracentrifuged at 100,000× *g* for 30 min at 4 °C. The resultant pellets were then reconstituted with 2% SDS and saved as SDS soluble fractions and stored at −80 °C. The protein concentrations were determined using the BCA assay (Takara Bio Inc., Shiga, Japan). Both RIPA1 and RIPA2 were subjected to Western blot and ELISA analysis, yielding similar results. Therefore, only the results of RIPA1 are shown in the figure. 

### 4.7. Immunoblots

Fractionated brain homogenates were dissolved in LDS sample buffer (Thermo Fisher Scientific, Waltham, MA, USA) containing 2%β-mercaptothanol. Proteins were separated in 10% Tris-Glycine gel and transferred to PVDF membranes. Membranes were blocked with 5% skim milk for 30 min at room temperature and incubated with PHF-1 antibody or an anti-human tau HJ8.5 antibody (a kind gift from Dr. David Holtzman) or an anti-GAPDH antibody (FUJIFILM Wako Pure Chemical Corporation, Osaka, Japan 016-25523) or eYFP antibody (Thermofisher Scientific, Waltham, MA, USA, A11122), followed by incubation with species-specific HRP-conjugated secondary antibodies. Bands were visualized using immunostar (FUJIFILM Wako Pure Chemical Corporation, Osaka, Japan), and signals were detected using ImageQuant LAS-4000 (GE Healthcare Japan, Tokyo, Japan). Bands were quantified using ImageQuant TL (GE Healthcare). Mice that exhibited injection failure, as determined via eYFP Western blot analysis, were omitted from the subsequent analysis.

### 4.8. Image Analysis

Immunostained brain sections were imaged using BZ-X710 (Keyence, Tokyo, Japan). Using ImageJ (v1.51k), we outputted the area of each brain region from tiled images and then binarized them to measure the proportion of positively stained areas by the PHF-1 antibody. In doing so, we set a threshold to detect diffuse staining patterns specifically observed during chronic stimulation and then measured the area of regions showing staining intensity above this threshold as the positively stained area. For each individual, we calculated the PHF1-positive area ratio by dividing the PHF1-positive area by the area of the respective region. The propagation index shown in Figure 2E was calculated based on the formula below.
$$Propagation\;index=\frac{PHF-1\;positive\;area\left(\%\right)\;in\;hippocampus}{PHF-1\;positive\;area\left(\%\right)\;in\;EC}\times 100$$

We defined the hippocampus in the brain section corresponding to AP (−2.1–2.4 mm) as the rostral hippocampus, and the hippocampus in the brain section corresponding to AP (−2.8–3.1 mm) as the caudal hippocampus, aligning them with the brain atlas. The entorhinal cortex in the brain section corresponding to AP (−4.0–4.3 mm) was defined as the medial entorhinal cortex. The PHF-1 immunostaining pattern in the rostral hippocampus was generally consistent with that in the caudal hippocampus, so only the staining images of the caudal hippocampus are included in the figures. For Western blot analysis, bands were quantified using ImageQuant TL (GE Healthcare).

### 4.9. Human-Specific Tau ELISA

Concentrations of human tau in brain homogenates were determined via sandwich ELISAs using Tau-5 as a coating antibody and biotinylated HT7 (Thermo Fisher Scientific, MN1000B) as a detection antibody, as described previously [40].

### 4.10. Statistics

All data were presented as mean ± S.E.M. Statistical analysis was performed using GraphPad Prism 9 (Graph Pad software, Boston, MA, USA). For the comparison between the eYFP group and the ChR2^C128S/D156A^ group in terms of immunohistochemical or biochemical analysis, as the equality of variances between the two groups could not be assumed, Welch’s *t*-test was employed. Statistically significant outliers (*p* > 0.05) were identified via Grubb’s test and omitted from further analysis.

## Figures and Tables

**Figure 1 ijms-25-09004-f001:**
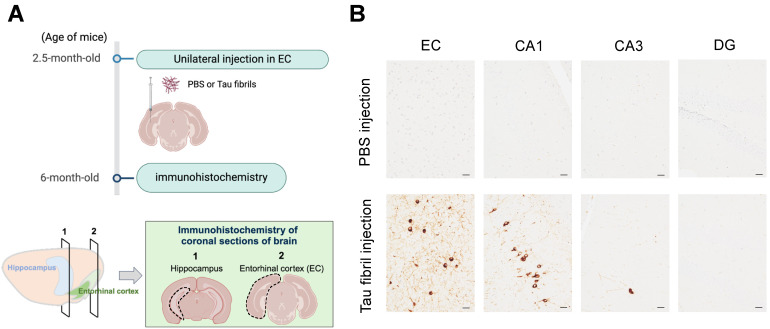
Seed-induced model of tau propagation from EC to hippocampus. (**A**) Schematic depiction of experimental design. Tau fibrils or PBS were unilaterally injected in the EC of 2.5-month-old PS19 mice. After 3.5 months post-injection, tau pathology in the ipsilateral coronal brain sections was analyzed via immunohistochemistry. Created with BioRender.com. (**B**) Representative images of PHF-1 staining in EC, CA1, CA3, and DG of the hippocampus following injection of tau fibrils in 6-month-old PS19 mice with hematoxylin staining. Scale bars, 20 μm.

**Figure 2 ijms-25-09004-f002:**
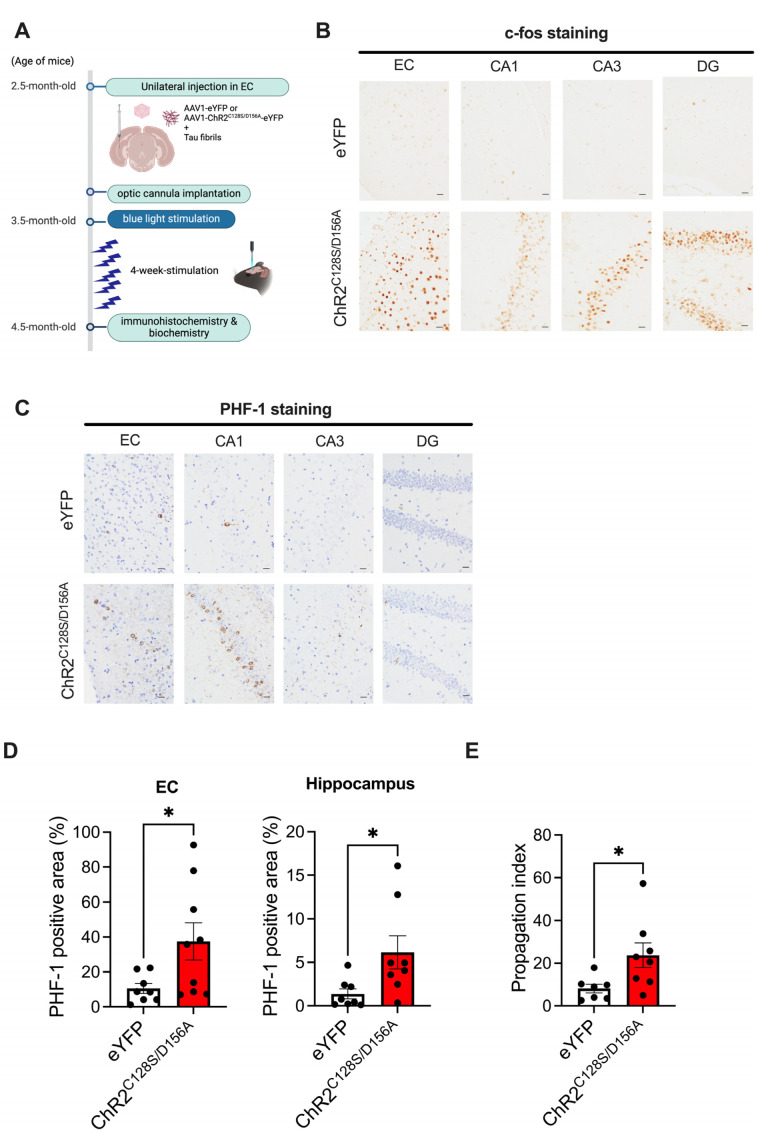
Chronic neuronal hyperexcitation exacerbates tau spreading from the EC to the hippocampus. (**A**) AAV1-expressing ChR2^C128S/D156A^ fused with eYFP or AAV1-expressing eYFP were injected into the EC of 2.5-month-old PS19 mice simultaneously with tau fibrils. Starting 4 weeks later, blue light stimulation was applied through optic cannulas implanted in the EC once a day for 4 weeks. At the end of the 4-week stimulation, when the mice reached 4.5 months of age, immunostaining for c-fos and p-tau (PHF-1) was performed. Created with BioRender.com. (**B**) Representative images of c-fos staining in EC, CA1, CA3 and DG of hippocampus following injection of AAV1 and tau fibrils in 4.5-month-old PS19 mice. Scale bars, 20 μm. (**C**) Representative images of PHF-1 staining in EC, CA1, CA3 and DG of hippocampus following injection of AAV1 and tau fibrils in 4.5-month-old PS19 mice with hematoxylin staining. Scale bars, 20 μm. (**D**) Quantification of % area covered by PHF-1 staining in PS19 mice that received AAV1 and tau fibrils. Unpaired *t* test with Welch’s correction. * *p* < 0.05. (**E**) Quantification of the ratio of tau accumulation in the hippocampus relative to tau accumulation in the EC for each mouse (propagation index). Unpaired *t* test with Welch’s correction. * *p* < 0.05.

**Figure 3 ijms-25-09004-f003:**
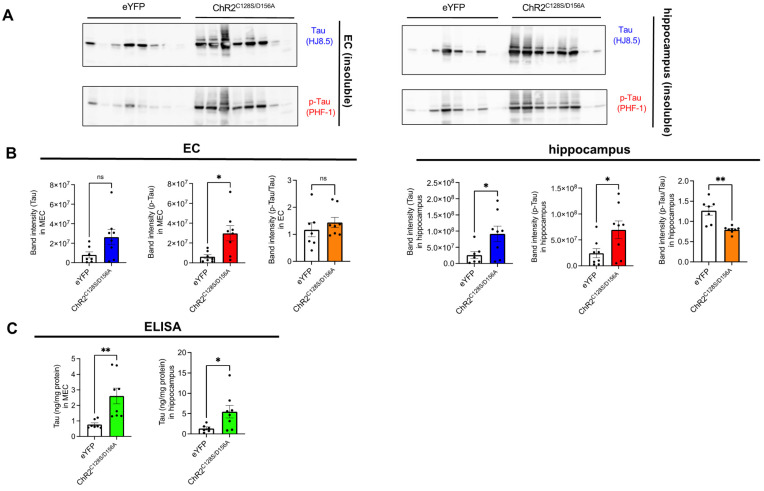
The levels of insoluble tau in EC and hippocampus after chronic neuronal hyperexcitation. (**A**) Immunoblots probing for human tau (HJ8.5) and p-tau (PHF-1) as well as p-tau ratio in insoluble fractions of EC or hippocampus of PS19 mice that received AAV1 and tau fibrils. (**B**) Quantification of immunoblots probing for human tau (HJ8.5) and p-tau (PHF-1) in insoluble fractions of EC or hippocampus of PS19 mice. Unpaired *t* test with Welch’s correction. * *p* < 0.05, ** *p* < 0.01, ns: Not significant. (**C**) Quantification of human tau in insoluble fractions of EC or hippocampus of PS19 mice by ELISA. Unpaired *t* test with Welch’s correction. * *p* < 0.05, ** *p* < 0.01.

**Figure 4 ijms-25-09004-f004:**
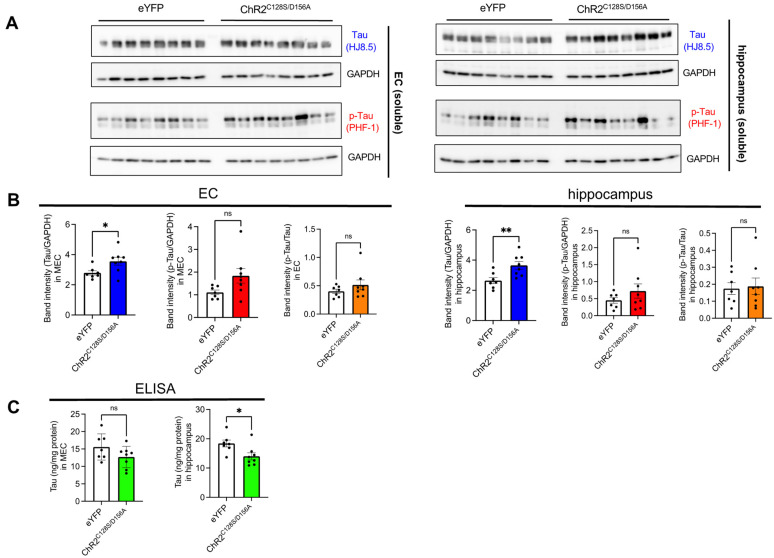
The levels of soluble tau in EC and hippocampus after chronic neuronal hyperexcitation (**A**) Immunoblots probing for human tau (HJ8.5) and p-tau (PHF-1) as well as p-tau ratio in soluble fractions of EC or hippocampus of PS19 mice that received AAV1 and tau fibrils. (**B**) Quantification of immunoblots probing for human tau (HJ8.5) and p-tau (PHF-1) in soluble fractions of EC or hippocampus of PS19 mice. Unpaired *t* test with Welch’s correction. * *p* < 0.05, ** *p* < 0.01. ns: Not significant. (**C**) Quantification of human tau in soluble fractions of EC or hippocampus of PS19 mice using ELISA. Unpaired *t* test with Welch’s correction. * *p* < 0.05. ns: Not significant.

## Data Availability

The data that support the findings of this study are available from the corresponding author, K.Y., upon reasonable request.

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
