# Peer review of "Chronic Neuronal Hyperexcitation Exacerbates Tau Propagation in a Mouse Model of Tauopathy"

_ijms, 2024, doi:10.3390/ijms25169004_

Round 1

Reviewer 1 Report

Comments and Suggestions for Authors

The authors examined whether Tau could spread via neural circuits to different brain regions due to neuronal hyperactivity. They found that chronic stimulation of neuronal connections between the entorhinal cortex and the hippocampus led to increased tau accumulation in both areas. This finding supports their hypothesis that abnormal neuronal activity promotes the spread of Tau, which may contribute to the progression of tauopathy. I have following comments.

1. Schematics need improvement to show unilateral injections. Figure 1A, is unclear at the moment. Please add scale bars to the images.

2. In Figure 2A, the schematic can be improved. Add scale bars to the images. Also, please mention in the results section that unilateral injections were made in the left hemisphere only. I found it quite interesting that control sections show zero c-fos labeled cells. I would like to know if there was any c-fos labeling in the contralateral hippocampus.

3. Please discuss which types of cells are c-fos positive and elaborate on the significance of using c-fos as a marker. Explain what c-fos is and why it is being used.

4. The authors are only using c-fos staining to measure the intrinsic brain activity. It's important for them to also conduct local field recordings or whole cell patch clamp recordings to demonstrate neuronal hyperactivity. If they are unable to do these experiments due to limited resources or time constraints, they should acknowledge this limitation in the discussion.

5. Citations are missing in the methods section.

Author Response

Reviewer 1:1. Schematics need improvement to show unilateral injections. Figure 1A, is unclear at the moment. Please add scale bars to the images.

Author response: We modified Figure 1 to show unilateral injection. We also added scale bars to the images.

Reviewer 1:2. In Figure 2A, the schematic can be improved. Add scale bars to the images. Also, please mention in the results section that unilateral injections were made in the left hemisphere only. I found it quite interesting that control sections show zero c-fos labeled cells. I would like to know if there was any c-fos labeling in the contralateral hippocampus.

Author response: We modified Figure 2 to show unilateral injection. We also added scale bars to the images. We now mention that unilateral injections were made in the left hemisphere only in the result sections. As we described in line 117 in page3, control sections also have c-fos labeling although signal intensity was quite low. We saw c-fos labeling in the contralateral hippocampus after chronic stimulation, which was now described in the result section along with the new figure (Figure S1).

Reviewer 1:3. Please discuss which types of cells are c-fos positive and elaborate on the significance of using c-fos as a marker. Explain what c-fos is and why it is being used.

Author response: We could not identify the cell type for all c-fos positive cells, but for those with a clear cell type, we described them in the paper. Additionally, we included an explanation of what c-fos is and why it is being used.

Reviewer 1:4. The authors are only using c-fos staining to measure the intrinsic brain activity. It's important for them to also conduct local field recordings or whole cell patch clamp recordings to demonstrate neuronal hyperactivity. If they are unable to do these experiments due to limited resources or time constraints, they should acknowledge this limitation in the discussion.

Author response: We thank this reviewer for the constructive comments. Unfortunately, we were not able to perform local field recordings or whole cell patch clamp recordings to demonstrate neuronal hyperactivity in ChR2-expressing PS19 mice due to limited resources or time constraints, we agree that the magnitude of neuronal activity elicited in our chronic stimulation paradigm should be ideally determined by electrical recording. We now mentioned this limitation in Discussion.

Reviewer 1:5. Citations are missing in the methods section.

Author response: We have added more citations in the method section in the revised manuscript.

Reviewer 2 Report

Comments and Suggestions for Authors

The presented paper entitled "Chronic Neuronal Hyperexcitation Exacerbates Tau Propaga- 2 tion in a Mouse Model of Tauopathy" by Nishida et al. bases its hypothesis on previous evidences about the effect of neuronal hyperexcitability as a factor able to enhance the propagation of proteins involved in neurodegeneration, in this case the Tau protein. The propagation of pathological Tau during tauopathy progression and in Alzheimer's disease (AD) is a relevant topic still under investigation and the clarification of the mechanisms involved is of great importance for the dementia field. In this paper the authors induce neuronal hyperexcitability in a mouse model for tauopathies by an optogenetic approach and injection of Tau seeds in the enthorinal cortex (EC). The authors observe an increase in Tau patholgical markers in the EC but also in the hippocampus, suggesting an activity-dependent propagation enhancement that could drive Tau pathology and AD progression.

The paper is well written and the data presented are clear and interesting. The experimental approach is technically solid and integrates different technical models to investigate the propagation of Tau pathology in mainly 2 brain areas primarally involved in AD. However, in my opinion the results reported are reliable but a bit preliminary and I believe that the paper could be easily emproved.

Major Comments:

- I have some concerns about the mouse model employed. The PS19 mouse model is based on a Tau mutation not associated with AD but with other tauopathies. Moreover this model usually presents a severe phenotype both in the cellular markers and in the behaviour. I understand you used this mouse model to enhance the effect of Tau pathology anda spreading but I think that you should explain better why to use this model and not others closer to an AD condition.

- In figure 1 it is reported the effect of Tau injection after 6 months while the following experiments are reported at 4.5 months, I understand that it is to validate your propagation model but please provide an explanation about why not to show the effect of chronic stimulation after 6 months.

- Do you see increased cell death in EC and hippocampus after neuronal stimulation?

- I believe that the results described would have a higher relevance by performing a behavioral test for memory to evaluate if the increased spreding of Tau also drives a functional damage. Moreover, pathological Tau in early AD stages seems to cause neuronal hyperexcitability (here some examples DOI10.1186/s13024-017-0176-x; https://doi.org/10.1016/j.jmb.2020.10.009),do you see an increase in seizures frequency after Tau treatments compared to controls?

- In the text there are several "data not shown" about control experiments, I suggest to add them as supplementary material.

- The discussion is clear but I suggest to add a brief comment about the possible mechanims involved in this propagation and how the Tau-dependent hyperexcitability itself could cause a positive feedback loop that further support this disequilibrium and propagation generated by the neuronal hyperexcitability (some references about this topic: DOI: 10.1186/s13024-017-0176-x; https://doi.org/10.1016/j.neurobiolaging.2020.06.004 https://doi.org/10.1016/j.jmb.2020.10.009; 10.3389/fcell.2023.1151223; https://doi.org/10.1016/j.neurot.2023.10.001; https://doi.org/10.1523/JNEUROSCI.2880-19.2020)

Minor Comments:

- Figure 2: the resolution seems lower compared to the other figures, please provide a better image

- Figure 2E: the values of the scale bar look too high if you compare the values between hippocampus and EC in figure 2D, could the authors clarify this point?

- Figure 4C: could the authors clarify the results compared to figure 4B?

- Do the authors see a difference based on sex for the propagation of Tau after chronic stimulation?

- In figure 3 and 4 the authors show an increase of p-Tau normalized on the housekeeping. I think that to see an increase in p-Tau it should be normilized on the housekeeping and the total Tau (normalized). Moreover, during AD progression an increase of total Tau is observed. Both soluble and insoluble fractions show higher Tau levels suggesting that after treatment and hyperexcitability you have an increase amount of total Tau (https://doi.org/10.1093/jnen/nlw105; https://doi.org/10.1186/s40478-016-0299-2). Please add total protein extracts and include also the controls without Tau injection, Moreover discuss this point.

- Figure 3A. Do you have loading controls? For example total protein (Ponceau or others)

Author Response

Reviewer2

Reviewer2: The paper is well written and the data presented are clear and interesting. The experimental approach is technically solid and integrates different technical models to investigate the propagation of Tau pathology in mainly 2 brain areas primarally involved in AD. However, in my opinion the results reported are reliable but a bit preliminary and I believe that the paper could be easily emproved.

Author response: We thank this reviewer for highly evaluating our manuscript.

Major Comments:

Reviewer2- I have some concerns about the mouse model employed. The PS19 mouse model is based on a Tau mutation not associated with AD but with other tauopathies. Moreover this model usually presents a severe phenotype both in the cellular markers and in the behaviour. I understand you used this mouse model to enhance the effect of Tau pathology anda spreading but I think that you should explain better why to use this model and not others closer to an AD condition.

Author response: A propagation model that involves injecting tau fibrils without mutations into wild-type mice or APP transgenic mice expressing endogenous tau has also been reported (Masuda-Suzukake et al., Brain comm 2020, He et al., Nat Med 2017). However, in many models, while tau fibrils induce tau accumulation at the injection site, tau propagation to distant brain regions is very limited compared to PS19 mice. Additionally, sufficient levels of propagation usually take more than six months in these models, which leaves concerns that chronic stimulation over the same period would significantly affect the mice's behavior. Therefore, we used PS19, which can induce propagation beyond the brain regions at an earlier stage. We mentioned this in line 78 in page 2.

Reviewer2- In figure 1 it is reported the effect of Tau injection after 6 months while the following experiments are reported at 4.5 months, I understand that it is to validate your propagation model but please provide an explanation about why not to show the effect of chronic stimulation after 6 months.

Author response: In the experiment shown in Figure 1, we found that massive tau propagation had occurred by the time the mice reached 6 months of age after fibril injection. Since further increases in tau propagation were expected with chronic stimulation, we chose 4.5 months of age to ensure that the effects of chronic stimulation on tau propagation would not be plateaued.

Reviewer2- Do you see increased cell death in EC and hippocampus after neuronal stimulation?

Author response: Severe atrophy accompanied by brain volume loss was not observed with chronic stimulation. However, we believe it is necessary to investigate whether cell death occurs with longer periods of chronic stimulation.

Reviewer2- I believe that the results described would have a higher relevance by performing a behavioral test for memory to evaluate if the increased spreding of Tau also drives a functional damage. Moreover, pathological Tau in early AD stages seems to cause neuronal hyperexcitability (here some examples DOI: 10.1186/s13024-017-0176-x; https://doi.org/10.1016/j.jmb.2020.10.009),do you see an increase in seizures frequency after Tau treatments compared to controls?

Author response: In our experiment, no obvious increase in seizure behavior was observed in the tau injection group compared to control. However, some reports suggest that it is soluble tau not aggregated tau that induces hyperexcitability, and we believe this cannot be verified in tau fibril injection experiments.

Reviewer2- In the text there are several "data not shown" about control experiments, I suggest to add them as supplementary material.

Author response: We appreciate the suggestion and now included the data as supplemental material.

Reviewer2- The discussion is clear but I suggest to add a brief comment about the possible mechanims involved in this propagation and how the Tau-dependent hyperexcitability itself could cause a positive feedback loop that further support this disequilibrium and propagation generated by the neuronal hyperexcitability (some references about this topic: DOI: 10.1186/s13024-017-0176-x; https://doi.org/10.1016/j.neurobiolaging.2020.06.004 https://doi.org/10.1016/j.jmb.2020.10.009; 10.3389/fcell.2023.1151223; https://doi.org/10.1016/j.neurot.2023.10.001; https://doi.org/10.1523/JNEUROSCI.2880-19.2020)

Author response: As this reviewer pointed out, it has been suggested a close relationship between tau and hyperexcitability. Our finding that prolonged neuronal hyperactivity promotes tau propagation suggests the existence of a positive feedback loop where tau-induced hyperexcitability further accelerates tau propagation. We discussed this point in the discussion section.

Minor Comments:

Reviewer2- Figure 2: the resolution seems lower compared to the other figures, please provide a better image

Author response: We now incorporated a image with better resolution.

Reviewer2- Figure 2E: the values of the scale bar look too high if you compare the values between hippocampus and EC in figure 2D, could the authors clarify this point?

Author response: The propagation index shown in Figure 2E was calculated based on the formula below.

Propagation index= x 100

It is the Hippocampus/EC ratio of tau accumulation in each individual mouse. Therefore, it represents different variances from tau accumulation in EC and HC themselves, which is reflected in the differences in the plots and error bars.

Reviewer2- Figure 4C: could the authors clarify the results compared to figure 4B?

Author response: Our interpretation of the data shown in Figure 4C is that it may reflect a potential decrease in monomeric tau and a concomitant increase in SDS-dissociable oligomeric tau. We added more words to make it clear at this point.

Reviewer2- Do the authors see a difference based on sex for the propagation of Tau after chronic stimulation?

Author response: In our experiment, no sex differences were observed in tau propagation. However, we believe that a larger number of mice is necessary to rigorously examine sex differences.

Reviewer2- In figure 3 and 4 the authors show an increase of p-Tau normalized on the housekeeping. I think that to see an increase in p-Tau it should be normilized on the housekeeping and the total Tau (normalized). Moreover, during AD progression an increase of total Tau is observed. Both soluble and insoluble fractions show higher Tau levels suggesting that after treatment and hyperexcitability you have an increase amount of total Tau (https://doi.org/10.1093/jnen/nlw105; https://doi.org/10.1186/s40478-016-0299-2). Please add total protein extracts and include also the controls without Tau injection, Moreover discuss this point.

Author response: We now normalized p-tau levels also by total tau and incorporated the data (p-tau ratio) in new Figure 3 and 4. Interestingly, when normalizing p-tau with total tau (p-tau ratio) in insoluble fractions, the value in hippocampus of the ChR2C128S/D156A group was found to be lower than that in the eYFP group although there was no changes in p-tau ratio in EC of either groups. This suggests that the insoluble tau propagated to hippocampus due to hyperexcitability has a relatively decreased level of phosphorylation.

In contrast, p-tau ratio in the soluble fraction remained unchanged between eYFP and ChR2C128S/D156A groups, suggesting that chronic neuronal stimulation did not solely facilitate phosphorylation of tau. 

Unfortunately, we were not able to measure tau levels in total extracts as brains collected after the chronic stimulation experiments all underwent serial extraction. Considering that the RIPA fraction contains the highest amount of tau, we expected that total tau would increase due to chronic stimulation. The control western blot data for PS19 mice without tau fibril injection was now included in supplemental material (Figure S3).

Reviewer2- Figure 3A. Do you have loading controls? For example total protein (Ponceau or others)

Author response: The SDS fraction shown in Figure 3 is a highly insoluble fraction obtained after two rounds of extractions with RIPA followed by extraction with 1M sucrose. Therefore, it is very challenging to normalize them with housekeeping proteins. Consequently, we quantified the tau in the SDS fraction not only by western blot but also by ELISA and normalized it to the wet tissue weight. Using these two different methods, we confirmed that the amount of tau in the SDS fraction was significantly increased in the ChR2 group.

Round 2

Reviewer 2 Report

Comments and Suggestions for Authors

I can appreciate that the authors improved the general quality of the paper by adding supporting figures and analyses and by clarifying important points of the study. I have just some minor concerns that I report below:

- The authors previously commented "Severe atrophy accompanied by brain volume loss was not observed with chronic stimulation". Due to the toxic effect of Tau on neuronal homeostasis and cell survival, in my opinion it would be relevant to add a behavioural test (for memory for example since hippocampus is significantly affected by Tau aggregation) and molecular or biochemical assays to show if cell death pathways are activated in the hippocampus (or inflammation pathways). I think that a functional experiment would strengthen your observations about the pathological relevance of Tau seeding propagation mediated by hyperexcitability.

- The authors declare " no obvious increase in seizure behavior was observed in the tau injection group compared to control". Please comment this observation in the text and add a supplementary graph or table with this result if available.

- In the manuscript the authors report their results mainly as representative of AD pathology and in fact they analyse the relationship between EC and hippocampus, key brain regions involved in AD progression. However, molecularly, the model employed PS19, bearing the TauP301S mutation, is not representative of AD but of other tauopathies and it is not discussed if this genetic background could influence the results obtained. I suggest to add a comment about Tau propagation mediated by hyperexcitability also in other tauopathies and to add the model employed as a possible limitation to represent an AD condition.
